# A Minimalist Approach to
# Offline Reinforcement Learning

**Scott Fujimoto**[1,2]  **Shixiang Shane Gu**[2]
[1]Mila, McGill University
[2]Google Research, Brain Team
scott.fujimoto@mail.mcgill.ca

## Abstract

Offline reinforcement learning (RL) defines the task of learning from a fixed batch of data. Due to errors in value estimation from out-of-distribution actions, most offline RL algorithms take the approach of constraining or regularizing the policy with the actions contained in the dataset. Built on pre-existing RL algorithms, modifications to make an RL algorithm work offline comes at the cost of additional complexity. Offline RL algorithms introduce new hyperparameters and often leverage secondary components such as generative models, while adjusting the underlying RL algorithm. In this paper we aim to make a deep RL algorithm work while making minimal changes. We find that we can match the performance of state-of-the-art offline RL algorithms by simply adding a behavior cloning term to the policy update of an online RL algorithm and normalizing the data. The resulting algorithm is a simple to implement and tune baseline, while more than halving the overall run time by removing the additional computational overheads of previous methods.

## 1   Introduction

Traditionally, reinforcement learning (RL) is thought of as a paradigm for online learning, where the interaction between the RL agent and its environment is of fundamental concern for how the agent learns. In offline RL (historically known as batch RL), the agent learns from a fixed-sized dataset, collected by some arbitrary and possibly unknown process [Lange et al., 2012]. Eliminating the need to interact with the environment is noteworthy as data collection can often be expensive, risky, or otherwise challenging, particularly in real-world applications. Consequently, offline RL enables the use of previously logged data or leveraging an expert, such as a human operator, without any of the risk associated with an untrained RL agent.

Unfortunately, the main benefit of offline RL, the lack of environment interaction, is also what makes it a challenging task. While most off-policy RL algorithms are applicable in the offline setting, they tend to under-perform due to "extrapolation error": an error in policy evaluation, where agents tend to poorly estimate the value of state-action pairs not contained in the dataset. This in turn affects policy improvement, where agents learn to prefer out-of-distribution actions whose value has been overestimated, resulting in poor performance [Fujimoto et al., 2019b]. The solution class for this problem revolves around the idea that the learned policy should be kept close to the data-generating process (or behavior policy), and has been given a variety of names (such as batch-constrained [Fujimoto et al., 2019b], KL-control [Jaques et al., 2019], behavior-regularized [Wu et al., 2019], or policy constraint [Levine et al., 2020]) depending on how this "closeness" is chosen to be implemented.

While there are many proposed approaches to offline RL, we remark that few are truly "simple", and even the algorithms which claim to work with minor additions to an underlying online RL

35th Conference on Neural Information Processing Systems (NeurIPS 2021).

algorithm make a significant number of implementation-level adjustments. In other cases, there are unmentioned hyperparameters, or secondary components, such as generative models, which make offline RL algorithms difficult to reproduce, and even more challenging to tune. Additionally, such mixture of details slow down the run times of the algorithms, and make causal attributions of performance gains and transfers of techniques across algorithms difficult, as in the case for many online RL algorithms [Henderson et al., 2017, Tucker et al., 2018, Engstrom et al., 2020, Andrychowicz et al., 2021, Furuta et al., 2021]. This motivates the need for more minimalist approaches in offline RL.

In this paper, we ask: *can we make a deep RL algorithm work offline with minimal changes?* We find that we can match the performance of state-of-the-art offline RL algorithms with a single adjustment to the policy update step of the TD3 algorithm [Fujimoto et al., 2018]. TD3's policy $\pi$ is updated with the deterministic policy gradient [Silver et al., 2014]:

$$\pi = \arg\max_{\pi} \mathbb{E}_{(s,a)\sim\mathcal{D}}[Q(s, \pi(s))]. \tag{1}$$

Our proposed change, TD3+BC, is to simply add a behavior cloning term to regularize the policy:

$$\pi = \arg\max_{\pi} \mathbb{E}_{(s,a)\sim\mathcal{D}}\left[\lambda Q(s, \pi(s)) - (\pi(s) - a)^2\right], \tag{2}$$

with a single hyperparameter $\lambda$ to control the strength of the regularizer. This modification can be made by adjusting only a single line of code. Additionally, we remark that normalizing the states over the dataset, such that they have mean $0$ and standard deviation $1$, improves the stability of the learned policy. Importantly, these are the *only* changes made to the underlying deep RL algorithm. To accommodate reproduciblity, all of our code is open-sourced[1].

We evaluate our minimal changes to the TD3 algorithm on the D4RL benchmark of continuous control tasks [Fu et al., 2020]. We find that our algorithm compares favorably against many offline RL algorithms, while being significantly easier to implement and more than halving the required computation cost. The surprising effectiveness of our minimalist approach suggests that in the context of offline RL, simpler approaches have been left underexplored in favor of more elaborate algorithmic contributions.

## 2   Related Work

Although to the best of our knowledge, we are the first to use TD3 with behavior cloning (BC) for the purpose of offline RL, we remark that combining RL with BC, and other imitation learning approaches, has been previously considered by many authors.

**RL + BC**. With the aim of accelerating reinforcement learning from examples (known as learning from demonstrations [Atkeson and Schaal, 1997]), BC has been used as a regularization for policy optimization with DDPG [Lillicrap et al., 2015, Nair et al., 2018, Goecks et al., 2020] and the natural policy gradient [Kakade, 2001, Rajeswaran et al., 2017], but with additional sophistication through modified replay buffers and pre-training. The most similar work to our own is a SAC+BC baseline [Haarnoja et al., 2018] from Nair et al. [2020] and an unpublished course project [Booher] combining PPO [Schulman et al., 2017] with BC.

**RL + Imitation**. Other than directly using BC with RL, imitation learning has been combined with RL in a variety of manners, such as mixed with adversarial methods [Zhu et al., 2018, Kang et al., 2018], used for pre-training [Pfeiffer et al., 2018], modifying the replay buffer [Večerík et al., 2017, Gulcehre et al., 2020], adjusting the value function [Kim et al., 2013, Hester et al., 2017], or reward shaping [Judah et al., 2014, Wu et al., 2021]. In all cases, these methods use demonstrations as a method for overcoming challenges in exploration or improving the learning speed of the RL agent.

**Offline RL**. As aforementioned, offline RL methods generally rely on some approach for "staying close" to the data. This may be implemented using an estimate of the behavior policy and then defining an explicit policy parameterization [Fujimoto et al., 2019b, Ghasemipour et al., 2020] or by using divergence regularization [Jaques et al., 2019, Kumar et al., 2019, Wu et al., 2019, Siegel et al., 2020, Guo et al., 2021, Kostrikov et al., 2021]. Other approaches use a weighted version of BC to favor high advantage actions [Wang et al., 2018, Peng et al., 2019, Siegel et al., 2020, Wang et al.,

---

[1]`https://github.com/sfujim/TD3_BC`

2020, Nair et al., 2020] or perform BC over an explicit subset of the data [Chen et al., 2020]. Some methods have modified the set of valid actions based on counts [Laroche et al., 2019] or the learned behavior policy [Fujimoto et al., 2019a]. Another direction is to implement divergence regularization as a form of pessimism into the value estimate [Nachum et al., 2019, Kumar et al., 2020, Buckman et al., 2020].

**Meta Analyses of RL Algorithms**. There are a substantial amount of meta analysis works on online RL algorithms. While some focus on inadequacies in the experimental protocols [Henderson et al., 2017, Osband et al., 2019], others study the roles of subtle implementation details in algorithms [Tucker et al., 2018, Engstrom et al., 2020, Andrychowicz et al., 2021, Furuta et al., 2021]. For example, Tucker et al. [2018], Engstrom et al. [2020] identified that superior performances of certain algorithms were more dependent on, or even accidentally due to, minor implementation rather than algorithmic differences. Furuta et al. [2021] study two broad families of off-policy algorithms, which most offline algorithms are based on, and find that a few subtle implementation details are strongly co-adapted and critical to specific algorithms, making attributions of performance gains difficult. Recent offline research also follows a similar trend, where a number of implementation modifications are necessary for high algorithmic performances (see Table 1). In contrast, we derive our algorithm by modifying the existing TD3 with only a few lines of codes. Our results suggest the community could also learn from careful explorations of simpler alternatives, besides emphasizing algorithmic novelties and complexities.

## 3 Background

**RL**. Reinforcement learning (RL) is a framework aimed to deal with tasks of sequential nature. Typically, the problem is defined by a Markov decision process (MDP) $(\mathcal{S}, \mathcal{A}, \mathcal{R}, p, \gamma)$, with state space $\mathcal{S}$, action space $\mathcal{A}$, scalar reward function $\mathcal{R}$, transition dynamics $p$, and discount factor $\gamma$ [Sutton and Barto, 1998]. The behavior of an RL agent is determined by a policy $\pi$ which maps states to actions (deterministic policy), or states to a probability distribution over actions (stochastic policy). The objective of an RL agent is to maximize the expected discounted return $\mathbb{E}_\pi[\sum_{t=0}^\infty \gamma^t r_{t+1}]$, which is the expected cumulative sum of rewards when following the policy in the MDP, where the importance of the horizon is determined by the discount factor. We measure this objective by a value function, which measures the expected discounted return after taking the action $a$ in state $s$: $Q^\pi(s, a) = \mathbb{E}_\pi[\sum_{t=0}^\infty \gamma^t r_{t+1} | s_0 = s, a_0 = a]$.

**BC**. Another approach for training policies is through imitation of an expert or behavior policy. Behavior cloning (BC) is an approach for imitation learning [Pomerleau, 1991], where the policy is trained with supervised learning to directly imitate the actions of a provided dataset. Unlike RL, this process is highly dependent on the performance of the data-collecting process.

**Offline RL**. Offline RL breaks the assumption that the agent can interact with the environment. Instead, the agent is provided a fixed dataset which has been collected by some unknown data-generating process (such as a collection of behavior policies). This setting may be considered more challenging as the agent loses the opportunity to explore the MDP according to its current beliefs, and instead from infer good behavior from only the provided data.

One challenge for offline RL is the problem of extrapolation error [Fujimoto et al., 2019b], which is generalization error in the approximate value function, induced by selecting actions not contained in the dataset. Simply put, it is difficult to evaluate the expected value of a policy which is sufficiently different from the behavior policy. Consequently, algorithms have taken the approach of constraining or regularizing the policy to stay near to the actions in the dataset [Levine et al., 2020].

## 4 Challenges in Offline RL

In this section, we identify key open challenges in offline RL through analyzing and evaluating prior algorithms. We believe these challenges highlight the importance of minimalist approaches, where performance can be easily attributed to algorithmic contributions, rather than entangled with the specifics of implementation.

**Implementation and Tuning Complexities**. RL algorithms are notoriously difficult to implement and tune [Henderson et al., 2017, Engstrom et al., 2020, Furuta et al., 2021], where minor code-level

|  | CQL [Kumar et al., 2020] | Fisher-BRC [Kostrikov et al., 2021] | TD3+BC (Ours) |
|---|---|---|---|
| Algorithmic Adjustments | Add regularizer to critic[†] Approximate logsumexp with sampling[‡] | Train a generative model[†‡] Replace critic with offset function Gradient penalty on offset function[†] | Add a BC term[†] |
| Implementation Adjustments | Architecture[†‡] Actor learning rate[†] Pre-training actor Remove SAC entropy term Max over sampled actions[‡] | Architecture[†‡] Reward bonus[†] Remove SAC entropy term | Normalize states |

Table 1: Implementation changes offline RL algorithms make to the underlying base RL algorithm. [†] corresponds to details that add additional hyperparameter(s), and [‡] corresponds to ones that add a computational cost.

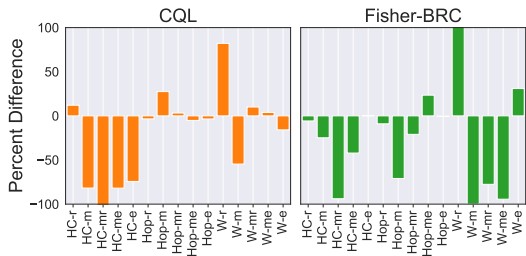

Figure 1: Percent difference of performance of offline RL algorithms and their simplified versions which remove implementation adjustments to their underlying algorithm. HC = HalfCheetah, Hop = Hopper, W = Walker, r = random, m = medium, mr = medium-replay, me = medium-expert, e = expert. Huge drops in performances show that the implementation complexities are crucial for achieving the best results in these prior algorithms.

optimizations and hyperparameters can have non-trivial impact of performance and stability. This problem may be additionally amplified in the context of offline RL, where evaluating changes to implementation and hyperparameters is counter-intuitive to the nature of offline RL, which explicitly aims to eliminate environment interactions [Paine et al., 2020, Yang et al., 2020].

Most offline RL algorithms are built explicitly on top of an existing off-policy deep RL algorithm, such as TD3 [Fujimoto et al., 2018] or SAC [Haarnoja et al., 2018], but then further modify the underlying algorithm with "non-algorithmic" implementation changes, such as modifications to network architecture, learning rates, or pre-training the actor network. We remark that a desirable property of an offline RL algorithm would be to minimally modify the underlying algorithm, so as to reduce the space of possible adjustments required to achieve a strong performance.

In Table 1 we examine the particular modifications made by two recent offline RL algorithms, CQL [Kumar et al., 2020] and Fisher-BRC [Kostrikov et al., 2021]. On top of algorithmic changes, CQL also adds a pre-training phase where the actor is only trained with imitation learning and selects the max action over a sampled set of actions from the policy during evaluation. Fisher-BRC adds a constant reward bonus to every transition. Both methods modify SAC by removing the entropy term in the target update and modify the default network architecture. These changes add supplementary hyperparameters which may need to be tuned or increase computational costs.

In the online setting, these changes are relatively inconsequential and could be validated with some simple experimentation. However, in the offline setting, where we cannot interact with the environment, making additional adjustments to the underlying algorithm should be considered as more costly as validating their effectiveness is no longer a trivial additional step. This is also problematic because unlike the algorithmic changes proposed by these papers, these implementation details are not well justified, meaning there is much less intuition as to when to include these details, or how to adjust them with minimal experimentation. In the case of the D4RL benchmark on MuJoCo tasks [Todorov et al., 2012, Fu et al., 2020], we have a strong prior that our base deep RL algorithm performs well, as SAC/TD3 are considered state-of-the-art (or nearly) in these domains. If additional changes are necessary, then it suggests the algorithmic contributions alone are insufficient.

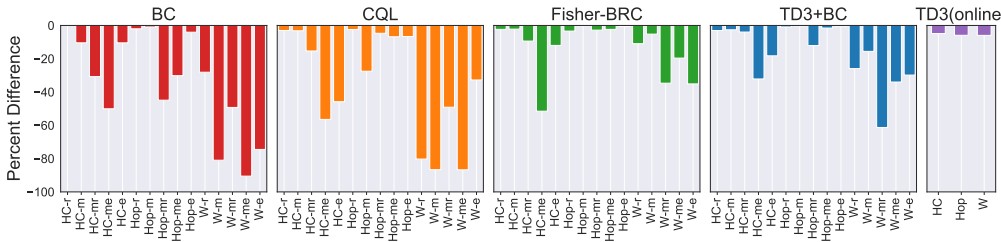

Figure 2: Percent difference of the worst episode during the 10 evaluation episodes at the last evaluation. This measures the deviations in performance at single point in time. HC = HalfCheetah, Hop = Hopper, W = Walker, r = random, m = medium, mr = medium-replay, me = medium-expert, e = expert. While online algorithms (TD3) typically have small episode variances per trained policy (as they should at *convergence*), all offline algorithms have surprisingly high episodic variances for *trained* policies.

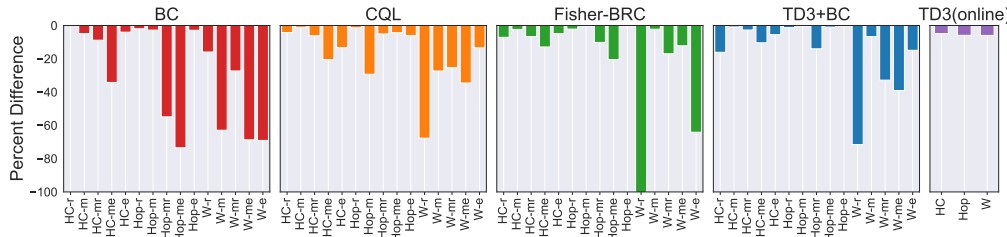

Figure 3: Percent difference of the worst evaluation during the last 10 evaluations. This measures the deviations in performance over a period of time. HC = HalfCheetah, Hop = Hopper, W = Walker, r = random, m = medium, mr = medium-replay, me = medium-expert, e = expert. Similarly to the result in Figure 2, all offline-trained policies have significant variances near the final stage of training that are absent in the online setting.

In Figure 1 we examine the percent difference in performance when removing the implementation changes in CQL and Fisher-BRC. There is a significant drop in performance in many of the tasks. This is not necessarily a death knell to these algorithms, as it is certainly possible these changes could be kept without tuning when attempting new datasets and domains. However, since neither paper make mention of a training/validation split, we can only assume these changes were made with their evaluation datasets in mind (D4RL, in this instance), and remark there is insufficient evidence that these changes may be universal. Ultimately, we make this point not to suggest a fundamental flaw with pre-existing algorithms, but to suggest that there should be a preference for making minimal adjustments to the underlying RL algorithm, to reduce the need for hyperparameter tuning.

**Extra Computation Requirement**. A secondary motivating factor for minimalism is avoiding the additional computational costs associated with modifying the underlying algorithm (in particular architecture) and more complex algorithmic ideas. In Table 3 (contained later in Section 6), we examine the run time of offline RL algorithms, as well as the change in cost over the underlying algorithm, and find their is a significant computational cost associated with these modifications. On top of the architecture changes, for CQL this is largely due to the costs of logsumexp over multiple sampled actions, and for Fisher-BRC, the costs associated with training the independent generative model. Since an objective of offline RL is to take advantage of existing, potentially large, datasets, there should be a preference for scalable and efficient solutions. Of course, run time should not come at the cost of performance, but as we will later demonstrate, there exists a simple, and computationally-free, approach for offline RL which matches the performance of current state-of-the-art algorithms.

**Instability of Trained Policies**. In analyzing the final trained policies of prior offline algorithms, we learned of a tangential, and open, challenge in the form of instability. In online RL, if the current policy is unsatisfactory, we can use checkpoints of previous iterations of the policy, or to simply continue training. However, in offline RL, the evaluation should only occur once by definition, greatly increasing the importance of the single policy at evaluation time. We highlight two versions of instability in offline RL. Figure 2 shows that in contrast to the online-trained policy, which converges to a robust low-variance policy, the offline-trained policy exhibits huge variances in performance

during a single evaluation. Therefore, even if the average performance is reasonable, the agent may still perform poorly on some episodes. Figure 3 shows instability over the set of evaluations, which means the performance of the agent may be dependent on the specific stopping point chosen for evaluation. This questions the empirical effectiveness of offline RL for safety-critical real-world use cases [Mandel et al., 2014, Gottesman et al., 2018, Gauci et al., 2018, Jaques et al., 2019, Matsushima et al., 2020] as well as the current trend of reporting only the mean-value of the final policy in offline benchmarking [Fu et al., 2020].

Such variances are not likely caused by high policy entropies (e.g. TD3+BC trains a deterministic policy), but rather our hypothesis is that such a problem is due to distributional shifts issues and poor generalizations across unobserved states caused by offline nature of training [Ross et al., 2011], where the optimized policy is never allowed to execute in the environment, similarly as in BC. This trait of offline RL algorithms appears to be consistent across all offline algorithms we evaluated, even for our minimalistic TD3+BC that is only a few lines change from TD3. While we could not solve this challenge sufficiently within the scope of this work, the fact that this is reproducible even in the minimalistic variant proves that this a fundamental problem shared by all offline training settings, and is a critical problem for the community to study in the future.

## 5 A Minimalist Offline RL Algorithm

A key problem in offline RL, extrapolation error, can be summarized as the inability to properly evaluate out-of-distribution actions. Consequently, there has been a variety of different approaches to limiting, or regularizing, action selection such that the learned policy is easier to evaluate with the given dataset. We remark that while minimizing say, KL divergence, is a both logical and valid approach for reducing extrapolation error, there is no fundamental argument why minimizing one divergence or distance metric should be better than another. Thus, rather than derive an entirely new approach, we focus on simplicity, and present an offline RL algorithm which requires minimal modifications to a pre-existing deep RL algorithm. As discussed in Section 4 a minimalist approach has a variety of benefits, such as reducing the number of hyperparameters to tune, increasing scalability by reducing computational costs, and providing an avenue for analyzing problems by disentangling algorithmic contributions from implementation details.

We now describe such an approach to offline RL. Our algorithm builds on top of TD3 [Fujimoto et al., 2018], making only two straightforward changes. Firstly, we add a behavior cloning regularization term to the standard policy update step of TD3, to push the policy towards favoring actions contained in the dataset $\mathcal{D}$:

$$\pi = \underset{\pi}{\arg\max}\, \mathbb{E}_{s \sim \mathcal{D}} \left[ Q(s, \pi(s)) \right] \rightarrow \pi = \underset{\pi}{\arg\max}\, \mathbb{E}_{(s,a) \sim \mathcal{D}} \left[ \lambda\, Q(s, \pi(s)) - (\pi(s) - a)^2 \right]. \quad (3)$$

Secondly, we normalize the features of every state in the provided dataset. Let $s_i$ be the $i$th feature of the state $s$, let $\mu_i\, \sigma_i$ be the mean and standard deviation, respectively, of the $i$th feature across the dataset:

$$s_i = \frac{s_i - \mu_i}{\sigma_i + \epsilon}, \quad (4)$$

where $\epsilon$ is a small normalization constant (we use $10^{-3}$). While we remark this is a commonly used implementation detail in many deep RL algorithms [Raffin et al., 2019], we highlight it as (1) we want complete transparency about all implementation changes and (2) normalizing provides a non-trivial performance benefit in offline RL, where it is particularly well-suited as the dataset remains fixed.

While the choice of $\lambda$ in Equation (3) is ultimately just a hyperparameter, we observe that the balance between RL (in maximizing $Q$) and imitation (in minimizing the BC term), is highly susceptible to the scale of $Q$. If we assume an action range of $[-1, 1]$, the BC term is at most 4, however the range of $Q$ will be a function of the scale of the reward. Consequently, we can add a normalization term into $\lambda$. Given the dataset of $N$ transitions $(s_i, a_i)$, we define the scalar $\lambda$ as:

$$\lambda = \frac{\alpha}{\frac{1}{N} \sum_{(s_i, a_i)} |Q(s_i, a_i)|}. \quad (5)$$

This is simply a normalization term based on the average absolute value of $Q$. In practice, we estimate this mean term over mini-batches, rather than the entire dataset. Although this term includes $Q$, it is

| | | BC | BRAC-p | AWAC | CQL | Fisher-BRC | TD3+BC |
|---|---|---|---|---|---|---|---|
| Random | HalfCheetah | 2.0 ±0.1 | 23.5 | 2.2 | 21.7 ±0.9 | 32.2 ±2.2 | 10.2 ±1.3 |
| | Hopper | 9.5 ±0.1 | 11.1 | 9.6 | 10.7 ±0.1 | 11.4 ±0.2 | 11.0 ±0.1 |
| | Walker2d | 1.2 ±0.2 | 0.8 | 5.1 | 2.7 ±1.2 | 0.6 ±0.6 | 1.4 ±1.6 |
| Medium | HalfCheetah | 36.6 ±0.6 | 44.0 | 37.4 | 37.2 ±0.3 | 41.3 ±0.5 | 42.8 ±0.3 |
| | Hopper | 30.0 ±0.5 | 31.2 | 72.0 | 44.2 ±10.8 | 99.4 ±0.4 | 99.5 ±1.0 |
| | Walker2d | 11.4 ±6.3 | 72.7 | 30.1 | 57.5 ±8.3 | 79.5 ±1.0 | 79.7 ±1.8 |
| Medium Replay | HalfCheetah | 34.7 ±1.8 | 45.6 | - | 41.9 ±1.1 | 43.3 ±0.9 | 43.3 ±0.5 |
| | Hopper | 19.7 ±5.9 | 0.7 | - | 28.6 ±0.9 | 35.6 ±2.5 | 31.4 ±3.0 |
| | Walker2d | 8.3 ±1.5 | -0.3 | - | 15.8 ±2.6 | 42.6 ±7.0 | 25.2 ±5.1 |
| Medium Expert | HalfCheetah | 67.6 ±13.2 | 43.8 | 36.8 | 27.1 ±3.9 | 96.1 ±9.5 | 97.9 ±4.4 |
| | Hopper | 89.6 ±27.6 | 1.1 | 80.9 | 111.4 ±1.2 | 90.6 ±43.3 | 112.2 ±0.2 |
| | Walker2d | 12.0 ±5.8 | -0.3 | 42.7 | 68.1 ±13.1 | 103.6 ±4.6 | 101.1 ±9.3 |
| Expert | HalfCheetah | 105.2 ±1.7 | 3.8 | 78.5 | 82.4 ±7.4 | 106.8 ±3.0 | 105.7 ±1.9 |
| | Hopper | 111.5 ±1.3 | 6.6 | 85.2 | 111.2 ±2.1 | 112.3 ±0.2 | 112.2 ±0.2 |
| | Walker2d | 56.0 ±24.9 | -0.2 | 57.0 | 103.8 ±7.6 | 79.9 ±32.4 | 105.7 ±2.7 |
| | Total | 595.3 ±91.5 | 284.1 | - | 764.3 ±61.5 | 974.6 ±108.3 | 979.3 ±33.4 |

Table 2: Average normalized score over the final 10 evaluations and 5 seeds. The highest performing scores are highlighted. CQL and Fisher-BRC are re-run using author-provided implementations to ensure an identical evaluation process, while BRAC and AWAC use previously reported results. ± captures the standard deviation over seeds. TD3+BC achieves effectively the same performances as the state-of-the-art Fisher-BRC, despite being much simpler to implement and tune and more than halving the computation cost.

not differentiated over, and is simply used to scale the loss. This formulation has the added benefit of normalizing the learning rate across tasks, as the gradient $\nabla_a Q(s, a)$ will also be dependent on the scale of $Q$. We use $\alpha = 2.5$ in our experiments.

This completes the description of TD3+BC. The Equations (3), (4), and (5) summarizes the entirety of our changes to TD3, and can be implemented by modifying only a handful of lines in most codebases.

## 6 Experiments

We evaluate our proposed approach on the D4RL benchmark of OpenAI gym MuJoCo tasks [Todorov et al., 2012, Brockman et al., 2016, Fu et al., 2020], which encompasses a variety of dataset settings and domains. Our offline RL baselines include two state-of-the-art algorithms, CQL [Kumar et al., 2020] and Fisher-BRC [Kostrikov et al., 2021], as well as BRAC [Wu et al., 2019] and AWAC [Nair et al., 2020] due to their algorithmic simplicity.

To ensure a fair and identical experimental evaluation across algorithms, we re-run the state-of-the-art algorithms CQL and Fisher-BRC using the author-provided implementations[2][3]. We train each algorithm for 1 million time steps and evaluate every 5000 time steps. Each evaluation consists of 10 episodes. Results for BRAC are obtained from the D4RL benchmark and from the CQL paper, while the AWAC results are obtained directly from the paper. BC results were obtained using our own implementation.

**D4RL.** We report the final performance results in Table 2 and display the learning curves in Figure 4. Although our method is very simplistic in nature, it surpasses, or matches, the performance of the current state-of-the-art offline RL algorithms in most tasks. Only Fisher-BRC exhibits a comparable

---

[2]`https://github.com/aviralkumar2907/CQL`
[3]`https://github.com/google-research/google-research/tree/master/fisher_brc`

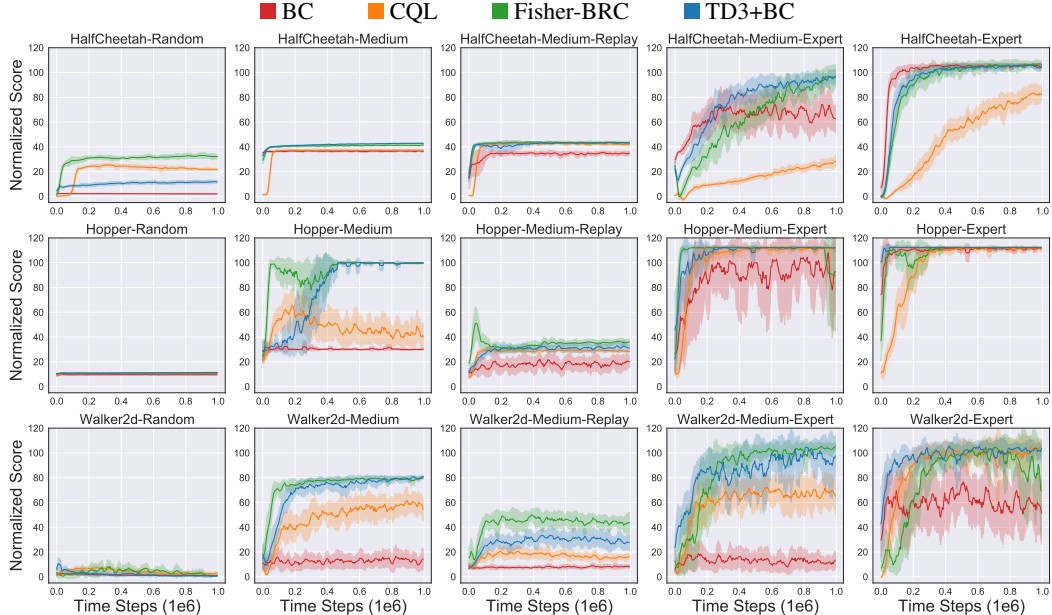

Figure 4: Learning curves comparing the performance of TD3+BC against offline RL baselines in the D4RL datasets. Curves are averaged over 5 seeds, with the shaded area representing the standard deviation across seeds. TD3+BC exhibits a similar learning speed and final performance as the state-of-the-art Fisher-BRC, without the need of pre-training a generative model.

| | CQL (GitHub) | Fisher-BRC (GitHub) | Fisher-BRC (Ours) | TD3+BC (Ours) |
|---|---|---|---|---|
| Implementation | 25m | 39m | 15m | < 1s |
| Algorithmic | 1h 29m | 33m | 58m | < 5s |
| Total | 4h 11m | 2h 12m | 2h 8m | 39m |

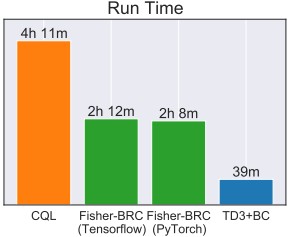

Table 3: Run time comparison of training each offline RL algorithm (does not include evaluation costs). (Left) Breakdown of the cost of the added implementation details (mainly architecture changes) and the algorithmic details by each method. (Right) Total training time of each algorithm. While CQL and Fisher-BRC have significantly increased computational costs over their base online RL algorithm due to various added complexities (e.g. see Table 1), TD3+BC has effectively no increase. This results in less than half of the computational cost of these prior state-of-the-art algorithms.

performance[4]. Examining the learning curves, we can see that our approach achieves a similar learning speed and stability, without requiring any pre-training phase.

**Run time.** We evaluate run time of training each of the offline RL algorithms for 1 million time steps, using the author-provoided implementations. Additionally, for fair comparison, we re-implement Fisher-BRC, allowing each method to be compared in the same framework (PyTorch [Paszke et al., 2019]). The results are reported in Table 3. Unsurprisingly, we find our approach compares favorably against previous methods in terms of wall-clock training time, effectively adding no cost to the underlying TD3 algorithm. All run time experiments were run with a single GeForce GTX 1080 GPU and an Intel Core i7-6700K CPU at 4.00GHz.

**Ablation.** In Figure 5, we perform an ablation study over the components in our method. As noted in previous work [Fujimoto et al., 2019b], without behavior cloning regularization, the RL algorithm alone is insufficient to achieve a high performance (except on some of the random data sets). We

---

[4]As noted by previous papers [Kostrikov et al., 2021], we remark the author-provided implementation of CQL achieves a lower result than their reported results.

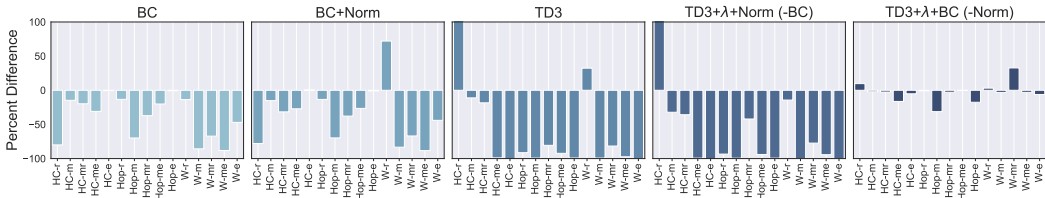

Figure 5: Percent difference of the performance of an ablation of our proposed approach, compared to the full algorithm. TD3+$\lambda$+BC+Norm refers to the complete algorithm, where Norm refers to the state feature normalization. HC = HalfCheetah, Hop = Hopper, W = Walker, r = random, m = medium, mr = medium-replay, me = medium-expert, e = expert. As expected, both BC and TD3 are necessary components to achieve a strong performance. While removing state normalization is not devastating to the performance of the algorithm, we remark it provides a boost in performance across many tasks, while being a straightforward addition.

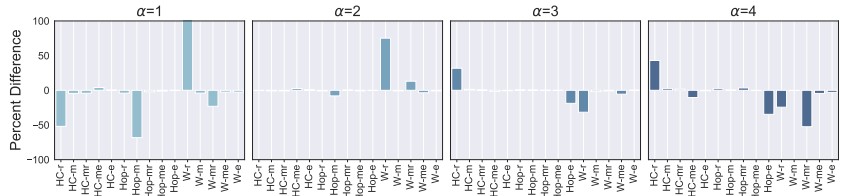

Figure 6: Percent difference of the performance of an ablation over $\alpha$, compared to the full algorithm. Recall the form of the sole hyperparameter $\lambda = \frac{\alpha}{\sum |Q(s,a)|}$, where $\lambda$ weights the $Q$ function during the policy update. $\alpha = 2.5$ is used by default. HC = HalfCheetah, Hop = Hopper, W = Walker, r = random, m = medium, mr = medium-replay, me = medium-expert, e = expert. While there is effectively no performance difference between $\alpha = 2$ and $\alpha = 3$, we remark that performance begins to degrade on select tasks as the algorithm begins to approach more RL ($\alpha = 4$) or imitation ($\alpha = 1$).

also note that our algorithm never underperforms vanilla behavior cloning, even on the expert tasks. Predictably, the removal of state normalization has the least significant impact, but it still provides some benefit across a range of tasks while being a minor adjustment. In Figure 6, we evaluate the sensitivity of the algorithm to the hyperparameter $\alpha$, where the weighting $\lambda = \frac{\alpha}{\sum_{(s,a)} |Q(s,a)|}$ on the value function is determined by $\alpha$. On many tasks, our approach is robust to this weighting factor, but note the performance on a subset of tasks begins to decrease as $\alpha$ begins to more heavily favor imitation ($\alpha = 1$) or RL ($\alpha = 4$).

## 7 Conclusion

Most recent advances in offline RL center around the idea of regularizing policy actions to be close to the support within batch data, and yet many state-of-the-art algorithms have significant complexities and additional modifications beyond base algorithms that lead to not only much slower run time, but also intractable attributions for sources of performance gains. Instead of complexity we optimize for simplicity, and introduce a minimalistic algorithm that achieves a state-of-the-art performance but is only a few lines of changes from the base TD3 algorithm, uses less than half of the computation time of competing algorithms, and has only one additional hyperparameter.

Additionally, we highlight existing open challenges in offline RL research, including not only the extra implementation, computation, and hyperparameter-tuning complexities that we successfully address in this work, but also call attention to the neglected problem of high episodic variance in offline-trained policies compared to online-trained (see Figures 2 and 3) that we as the community should address in future works and benchmarking. Finally, we believe the sheer simplicity of our approach highlights a possible overemphasis on algorithmic complexity made by the community, and we hope to inspire future work to revisit simpler alternatives which may have been overlooked.

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
