# A    Broader Impact

Offline RL will have societal impact by enabling new applications for reinforcement learning which can benefit from offline logged data such as robotics or healthcare applications, where collecting data is difficult, time-gated, expensive, etc. This may include potentially negative applications such as enforcing addictive behavior on social media. Another limitation to offline RL is that it is subject to any biases contained in the data set and can influenced by the data-generating policy.

For our specific algorithm, TD3+BC, given the performance gain over existing state-of-the-art methods is minimal, it would be surprising to see our paper result in significant impact in these contexts. However, where we might see impact is in enabling new users access to offline RL by reducing the computational cost, or burden of implementation, from having a simpler approach to offline RL. In other words, we foresee the impact our work is in accessibility and ease-of-use, and not through changing the scope of possible applications.

# B    Experimental Details

**Software**. We use the following software versions:

- Python 3.6
- Pytorch 1.4.0 [Paszke et al., 2019]
- Tensorflow 2.4.0 [Abadi et al., 2016]
- Gym 0.17.0 [Brockman et al., 2016]
- MuJoCo 1.50[1] [Todorov et al., 2012]
- mujoco-py 1.50.1.1

All D4RL datasets [Fu et al., 2020] use the v0 version.

**Hyperparameters**. Our implementations of TD3[2] [Fujimoto et al., 2018], CQL[3] [Kumar et al., 2020], and Fisher-BRC[4] [Kostrikov et al., 2021] are based off of their respective author-provided implementations from GitHub. For TD3+BC, only $\alpha$ was tuned in the range $(1, 2, 2.5, 3, 4)$ on Hopper-medium-v0 and Hopper-expert-v0 on a single seed which was unused in final reported results. We use default hyperparameters according to each GitHub whenever possible. For CQL we modify the GitHub defaults for the actor learning rate and use a fixed $\alpha$ rather than the Lagrange variant, matching the hyperparameters defined in their paper (which differs from the GitHub), as we found the original hyperparameters performed better. Our re-implementation of Fisher-BRC (in PyTorch rather than Tensorflow) is used only for run time experiments.

We outline the hyperparameters used by TD3+BC in Table 1, CQL in Table 2, and Fisher-BRC in Table 3.

**Heuristics for selecting** $\lambda$. While we find a single setting of $\lambda$ works across all datasets, some practitioners may be interested in guidelines in choosing $\lambda$, such as setting it to a fixed constant. Note the aim in our heuristic of normalizing by the average absolute value is to balance the importance of value maximization and behavior cloning. With domain knowledge, this heuristic can be circumvented by selecting $\lambda$ roughly equal to $\alpha$ over the expected average value. We can also choose $\lambda$ by considering the value estimate of the agent– if we see divergence in the value function due to extrapolation error [Fujimoto et al., 2019], then we need to decrease $\lambda$ such that the BC term is weighted more highly. Alternatively, if the performance resembles the performance of the behavior agent, then higher values of $\lambda$ should be considered.

---

[1]License information: `https://www.roboti.us/license.html`

[2]`https://github.com/sfujim/TD3`, commit 6a9f76101058d674518018ffbb532f5a652c1d37

[3]`https://github.com/aviralkumar2907/CQL`, commit d67dbe9cf5d2b96e3b462b6146f249b3d6569796

[4]`https://github.com/google-research/google-research/tree/master/fisher_brc`, commit 9898bb462a79e727ca5f413dba0ac3c4ee48d6c0

| | Hyperparameter | Value |
|---|---|---|
| | Optimizer | Adam [Kingma and Ba, 2014] |
| | Critic learning rate | 3e-4 |
| | Actor learning rate | 3e-4 |
| | Mini-batch size | 256 |
| TD3 Hyperparameters | Discount factor | 0.99 |
| | Target update rate | 5e-3 |
| | Policy noise | 0.2 |
| | Policy noise clipping | (-0.5, 0.5) |
| | Policy update frequency | 2 |
| | Critic hidden dim | 256 |
| | Critic hidden layers | 2 |
| Architecture | Critic activation function | ReLU |
| | Actor hidden dim | 256 |
| | Actor hidden layers | 2 |
| | Actor activation function | ReLU |
| TD3+BC Hyperparameters | $\alpha$ | 2.5 |

Table 1: TD3+BC Hyperparameters. Recall the form of $\lambda = \frac{\alpha}{\frac{1}{N}\sum_{(s,a)}|Q(s,a)|}$. The hyperparameters of TD3 are not modified from the TD3 GitHub.

| | Hyperparameter | Value |
|---|---|---|
| | Optimizer | Adam [Kingma and Ba, 2014] |
| | Critic learning rate | 3e-4 |
| | Actor learning rate | 3e-5[†] |
| SAC Hyperparameters | Mini-batch size | 256 |
| | Discount factor | 0.99 |
| | Target update rate | 5e-3 |
| | Target entropy | -1 · Action Dim |
| | Entropy in Q target | False[†] |
| | Critic hidden dim | 256 |
| | Critic hidden layers | 3[†] |
| Architecture | Critic activation function | ReLU |
| | Actor hidden dim | 256 |
| | Actor hidden layers | 3[†] |
| | Actor activation function | ReLU |
| | Lagrange | False |
| | $\alpha$ | 10 |
| CQL Hyperparameters | Pre-training steps | 40e3 |
| | Num sampled actions (during eval) | 10 |
| | Num sampled actions (logsumexp) | 10 |

Table 2: CQL Hyperparameters. We use the hyperparameters defined in the CQL paper rather than the default settings in the CQL GitHub as we found those settings performed poorly. [†] denotes hyperparameters which deviate from the original SAC hyperparameters.

|  | Hyperparameter | Value |
|---|---|---|
| | Optimizer | Adam [Kingma and Ba, 2014] |
| | Critic Learning Rate | 3e-4 |
| | Actor Learning Rate | 3e-4 |
| SAC Hyperparameters | Mini-batch size | 256 |
| | Discount factor | 0.99 |
| | Target update rate | 5e-3 |
| | Target entropy | -1 · Action Dim |
| | Entropy in Q target | False[†] |
| | Critic hidden dim | 256 |
| | Critic hidden layers | 3[†] |
| Architecture | Critic activation function | ReLU |
| | Actor hidden dim | 256 |
| | Actor hidden layers | 3[†] |
| | Actor activation function | ReLU |
| | Num Gaussians | 5 |
| Generative Model | Optimizer | Adam [Kingma and Ba, 2014] |
| Hyperparameters | Learning rate | (1e-3, 1e-4, 1e-5) |
| | Learning rate schedule | Piecewise linear (0, 8e5, 9e5) |
| | Target entropy | -1 · Action Dim |
| | Hidden dim | 256 |
| Generative Model Architecture | Hidden layers | 2 |
| | Activation function | ReLU |
| Fisher-BRC Hyperparameters | Gradient penalty $\lambda$ | 0.1 |
| | Reward bonus | 5 |

Table 3: Fisher-BRC Hyperparameters. We use the default hyperparameters in the Fisher-BRC GitHub. [†] denotes hyperparameters which deviate from the original SAC hyperparameters.

## C Additional Experiments

### C.1 Additional Datasets

A concern of TD3+BC is the poor performance on random data. In Table 4 we mix the random and expert datasets from D4RL [Fu et al., 2020], by randomly selecting half the transitions from each dataset and concatenating. We find that TD3+BC performs comparatively to Fisher-BRC. However, both algorithms underperform CQL on Walker2d. One hypothesis for this performance gap is due to the poor performance of BC on the Walker2d expert dataset (see Table 2 and Figure 4 in the main body). For completeness we also report the performance of TD3+BC on the D4RL AntMaze datasets. For this domain we found that state feature normalization was harmful to performance and was not included. All other hyperparameters remain unchanged.

| | CQL | Fisher-BRC | TD3+BC |
|---|---|---|---|
| HalfCheetah | 73.3$\pm$6.9 | 105.8$\pm$2.5 | 101.9$\pm$7.6 |
| Hopper | 110.8$\pm$2.4 | 111.9$\pm$0.9 | 112.2$\pm$0.3 |
| Walker2d | 100.3$\pm$8.5 | 32.0$\pm$37.3 | 28.8$\pm$23.4 |

Table 4: Average normalized score over the final 10 evaluations and 5 seeds on a mixture of 50% of the random D4RL dataset and 50% of the expert D4Rl dataset. CQL and Fisher-BRC are re-run using author-provided implementations to ensure an identical evaluation process. $\pm$ captures the standard deviation over seeds.

| | TD3+BC |
|---|---|
| AntMaze-Umaze | 78.6$\pm$33.3 |
| AntMaze-Umaze-Diverse | 71.4$\pm$20.7 |
| AntMaze-Medium-Diverse | 10.6$\pm$10.1 |
| AntMaze-Medium-Play | 3.0$\pm$4.8 |
| AntMaze-Large-Diverse | 0.2$\pm$0.4 |
| AntMaze-Large-Play | 0.0$\pm$0.0 |

Table 5: Average normalized score over the final 10 evaluations and 5 seeds on the AntMaze environments. $\pm$ captures the standard deviation over seeds.

### C.2 State Feature Normalization with Other Algorithms

To better understand the effectiveness of state feature normalization, we apply it to both CQL and Fisher-BRC. We report the percent difference of including this technique in Figure 1. We find that this generally has minimal impact on performance.

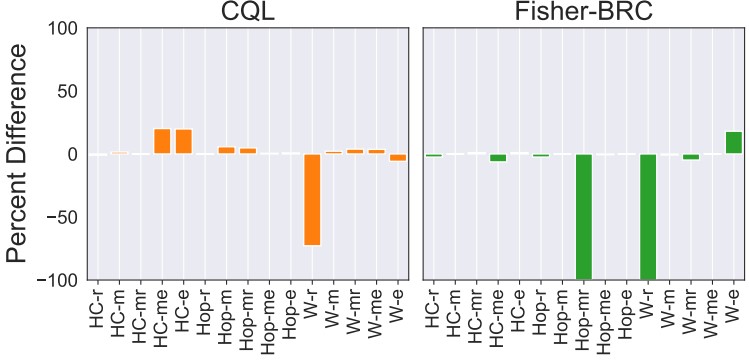

Figure 1: Percent difference of performance of offline RL algorithms when adding normalization to state features. HC = HalfCheetah, Hop = Hopper, W = Walker, r = random, m = medium, mr = medium-replay, me = medium-expert, e = expert.

## C.3 Benchmarking against the Decision Transformer

The Decision Transformer (DT) is concurrent work which examines the use of the transformer architecture in offline RL, by framing the problem as sequence modeling [Chen et al., 2021]. The authors use the D4RL -v2 datasets, which non-trivially affects performance and make direct comparison to results on the -v0 datasets inaccurate. To compare against this approach, we re-run TD3+BC on the -v2 datasets. Results are reported in Table 6. Although DT uses state of the art techniques from language modeling, and some per-environment hyperparameters, we find TD3+BC achieves a similar performance without any additional hyperparameter tuning. Furthermore, we benchmark the training time of DT against TD3+BC, using the author-provided implementation[5] in Figure 2, using the same experimental setup; a single GeForce GTX 1080 GPU and an Intel Core i7-6700K CPU at 4.00GHz. Unsurprisingly, TD3+BC trains significantly faster as it does not rely on the expensive transformer architecture.

| Dataset | Environment | DT | TD3+BC |
|---|---|---|---|
| Random | HalfCheetah | - | $11.0 \pm 1.1$ |
| | Hopper | - | $8.5 \pm 0.6$ |
| | Walker2d | - | $1.6 \pm 1.7$ |
| Medium | HalfCheetah | $42.6 \pm 0.1$ | $48.3 \pm 0.3$ |
| | Hopper | $67.6 \pm 1$ | $59.3 \pm 4.2$ |
| | Walker2d | $74.0 \pm 1.4$ | $83.7 \pm 2.1$ |
| Medium-Replay | HalfCheetah | $36.6 \pm 0.8$ | $44.6 \pm 0.5$ |
| | Hopper | $82.7 \pm 7$ | $60.9 \pm 18.8$ |
| | Walker2d | $66.6 \pm 3$ | $81.8 \pm 5.5$ |
| Medium-Expert | HalfCheetah | $86.8 \pm 1.3$ | $90.7 \pm 4.3$ |
| | Hopper | $107.6 \pm 1.8$ | $98.0 \pm 9.4$ |
| | Walker2d | $108.1 \pm 0.2$ | $110.1 \pm 0.5$ |
| Expert | HalfCheetah | - | $96.7 \pm 1.1$ |
| | Hopper | - | $107.8 \pm 7$ |
| | Walker2d | - | $110.2 \pm 0.3$ |
| | Total (DT) | $672.6 \pm 16.6$ | $677.4 \pm 45.6$ |
| | Total | - | $1013.2 \pm 57.4$ |

Table 6: Average normalized score using the D4RL -v2 datasets. The highest performing scores are highlighted. $\pm$ captures the standard deviation over seeds. Total (DT) sums scores over the subset of tasks with DT results. TD3+BC results are taken following the same experimental procedure as the -v0 datasets, averaging over the final 10 evaluations and 5 seeds. No additional hyperparameter tuning was performed. DT results are taken directly from the paper and uses 3 seeds. TD3+BC and DT achieve a comparable performance.

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

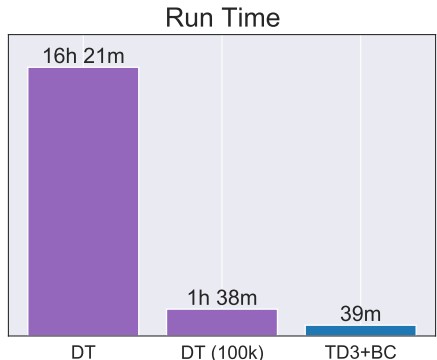

Figure 2: Benchmarking wall-clock training time of DT and TD3+BC over 1 million steps. Does not include evaluation costs. We remark that the DT was trained with only 100k time steps, which reduces the computational cost substantially, but even with this reduction, the DT takes over twice as long as TD3+BC to train. For many of the D4RL tasks the performance of TD3+BC converges before 100k time steps (see the learning curves in Figure 4 of the main body), but unlike the DT, we can let TD3+BC run for the full 1 million steps without incurring significant computational costs.

Justin Fu, Aviral Kumar, Ofir Nachum, George Tucker, and Sergey Levine. D4rl: Datasets for deep data-driven reinforcement learning. 2020.

Scott Fujimoto, Herke van Hoof, and David Meger. Addressing function approximation error in actor-critic methods. In *International Conference on Machine Learning*, volume 80, pages 1587–1596. PMLR, 2018.

Scott Fujimoto, David Meger, and Doina Precup. Off-policy deep reinforcement learning without exploration. In *International Conference on Machine Learning*, pages 2052–2062, 2019.

Diederik Kingma and Jimmy Ba. Adam: A method for stochastic optimization. *arXiv preprint arXiv:1412.6980*, 2014.

Ilya Kostrikov, Jonathan Tompson, Rob Fergus, and Ofir Nachum. Offline reinforcement learning with fisher divergence critic regularization. *arXiv preprint arXiv:2103.08050*, 2021.

Aviral Kumar, Aurick Zhou, George Tucker, and Sergey Levine. Conservative q-learning for offline reinforcement learning. *arXiv preprint arXiv:2006.04779*, 2020.

Adam Paszke, Sam Gross, Francisco Massa, Adam Lerer, James Bradbury, Gregory Chanan, Trevor Killeen, Zeming Lin, Natalia Gimelshein, Luca Antiga, et al. Pytorch: An imperative style, high-performance deep learning library. In *Advances in Neural Information Processing Systems*, pages 8024–8035, 2019.

Emanuel Todorov, Tom Erez, and Yuval Tassa. Mujoco: A physics engine for model-based control. In *IEEE/RSJ International Conference on Intelligent Robots and Systems (IROS)*, pages 5026–5033. IEEE, 2012.