# OpenReview forum: "A Minimalist Approach to Offline Reinforcement Learning"
_NeurIPS.cc/2021/Conference — NeurIPS 2021 Spotlight_

### Official Review · Reviewer_H8Cu · 2021-07-12

**Rating:** 7
**Confidence:** 5

**Summary:**

This paper proposes a new, simple approach to offline reinforcement learning, based on two modifications on top of a standard online RL algorithm (TD3) - introducing a BC loss to the policy loss, and normalizing the state features in the datasets. The authors show that these modifications achieve state-of-the-art performance for offline RL in the MuJoCo domain, even when compared to prior methods which involve many more moving pieces.

**Limitations And Societal Impact:**

Yes, the authors discuss some limitations of the method with regards to stability of the learned policies, and potential negative consequences such as negative applications of offline RL and data bias.

**Main Review:**

The originality of the work is small, as the proposed method is a small change on top of prior work. However, the authors recognize this, and I can see that the proposed method (TD3+BC) will be a strong but simple to implement baseline for all future offline RL algorithms going forward, so I view the contribution of this paper as potentially valuable and significant. The fact that such a simple modification results in near state-of-the-art results is surprising and potentally means there is still a lot of work for improving offline RL algorithms going forward.

The writing of the paper is good and easy to understand. It is clear what the motivation of the paper (to propose a simple, easy-to-implement but competitive offline RL method) and I believe that the work meets those claims.

For experiments, I believe that among model-free methods, these are the strongest or close to state-of-the-art results, which is fairly surprising given the simplicity of the method. One concern I have with the results is that they are primarily done in the MuJoCo domain, which are some of the easier tasks contained within the D4RL benchmark. I would be curious to see what the performance of the method is on a task such as the AntMaze navigation task, since in principle the task requires one to use dynamic programming to stitch together multiple incomplete trajectories. This would be a more interesting excercise for an offline RL algorithm where BC would not likely be as beneficial as it is for the MuJoCo tasks.

Overall, I think this is a good paper, mainly on the premise that it a) seems to be an overlooked method by the community, and b) achieves state-of-the-art results by a small margin on the domains reported. I think the paper could be stronger if there was more analysis of the method given that it is so simple (such as a theoretical analysis of the modified objective - does it still result in a convergent algorithm in simplified cases such as tabular, does adding BC loss prevent the method from achieving optimal returns, etc.). However, I still think the current experimental results are in general satisfactory.



**Time Spent Reviewing:**

1

---

> ### Author Response · Authors · 2021-08-10
> **Response to Reviewer H8Cu**
>
> Thank you for your review & comments. We’re glad that the main motivation/contributions of the paper came across clearly. We agree with your assessment that our work really highlights the fact that offline RL algorithms can still be greatly improved. We will take in consideration your suggested improvements to the paper for the final draft.

---

### Official Review · Reviewer_6fNL · 2021-07-16

**Rating:** 7
**Confidence:** 4

**Summary:**

The authors revisit addition of a behavior regularization term for offline reinforcement learning but with minimal change to existing RL approach, namely TD3. They showcase their approach in D4RL gym environments. The results are surprisingly good for such a small change and  are competitive with SOTA for the most part.


**Ethics Review Area:**

["I don’t know"]

**Limitations And Societal Impact:**

Yes

**Main Review:**


**Strengths:**

S1: Overall, the proposed method is very straightforward, fairly intuitive, and the performance seems comparable with the presented baselines.

S3: The experimental results present a compelling case for simplicity in offline RL and calls into attention the problem of high variance in offline-trained policies.

S2: With a few exceptions (discussed below), the presentation quality is high. Figures are and tables clear/readable. Writing is clear.

S3: The codebase accompanying the publication is minimal by design and supports high reproducibility.

**Weaknesses:**

W1: No free Lunch - The proposed algorithm while being simple seems vulnerable to noisy datasets with a large number of sub-optimal trajectories. It would be nice to see some experiments where for the given datasets of  "random" and "replay" tags are combined with "expert" datasets. This will reveal the effect of regularization in a dataset collected from  diverse set of sub-optimal policies. Datasets with noisy action spaces would also be an interesting addition. The paper fails to showcase these edge cases where the current simple algorithm might fail over the more complex offline RL approaches.

W2: The choice of hyperparameter does seem to affect the performance of the algorithm, However the initial choice of 2.5 seems to be arbitrary. This is because even if the actions are normalized the regularization term is dependent on dimension of the action space. Moreover a simple addition of random dimension in the action space might affect the performance of the solution. Hence it is still not clear how to heuristically choose the value of lambda for a new problem.


**Minor Clarity Issues/ Comments:**

CL1: It would also be interesting to see if the performance holds up for challenging tasks such as Ant-Maze.

CL2: In figure 5 It would be easy for the reader if it is explicitly mentioned that the full algorithm equates to" TD3 + BC + NORM + lambda" and replacing "- Norm" with " TD3 + BC + lambda". (If I understood the figure correctly.)

**Time Spent Reviewing:**

6

---

> ### Author Response · Authors · 2021-08-10
> **Response to Reviewer 6fNL**
>
> Thank you for your review & comments. To respond/comment on your listed weaknesses:
>
> W1: We agree that our method is likely to suffer when exposed to suboptimal trajectories (or a high ratio of suboptimal trajectories) due to the nature of the approach of staying close to the data. Consequently, this is also a problem with related approaches such as BCQ, BRAC, and Fisher-BRC. We will add experiments which analyze this weakness in more detail (such as the proposed experiment mixing random/replay with expert data) in the final draft.
>
> W2: We agree that the necessity of choosing the hyperparameter is a weakness of the method. However we do want to remark that:
> 1.	We find a value (2.5) which performs well across 15 datasets in D4RL, which suggests that the choice of hyperparameter is not overly sensitive to changes in data distribution and task.
> 2.	This is ultimately a weakness of all existing offline RL methods, as they all must choose a level of conservatism; the benefit of our approach is that this hyperparameter is arguably more interpretable and there is only a single hyperparameter which needs to be tuned unlike other methods (for example, see our Table 1).
>
> That being said, this is a fair criticism of the method, and offline hyperparameter tuning is an important direction for the field.
>
> CL2: Your interpretation of Figure 5 is correct, and we will adjust the title accordingly, thanks for pointing this out.

---

> > ### Comment · Reviewer_6fNL · 2021-08-22
> > **Following up.**
> >
> > Thank you for the response.
> >
> > [W1] I look forward to the new experiments and its results.
> > [W2] While I do agree that the algorithm does not seem to be overly sensitive to the hyper-parameter, I still believe some additional comments regarding heuristics/guideline for hyper-parameters would be important. Is the algorithm robust even towards an order of magnitude change in the hyper-parameter?

---

> > > ### Author Response · Authors · 2021-08-24
> > > **On hyperparameter selection**
> > >
> > > Ultimately the choice of hyperparameter is around balancing maximizing the value function and the BC term, which means the value should need to be within an order of magnitude around 1, or one term will dominate. There is, for example, the option to directly learn the hyperparameter such that both terms (Q and BC) get optimized but we chose not to go this route because (1) we wanted a simpler algorithm and (2) we did not find it necessary since we had a single value that worked well across all D4RL tasks. We can add a short discussion to the paper on some strategies for choosing the hyperparameter if the default value is ineffective. Thank you for your input!

---

### Official Review · Reviewer_63kT · 2021-07-19

**Rating:** 5
**Confidence:** 4

**Summary:**

This paper presents a simple offline RL method that only requires a few lines of changes in code. The authors add a BC loss as the regularization to the standard policy update objective while also normalizing the states. The authors also normalize the coefficient of the BC loss with the average Q values in the dataset to balance the policy update objective and the BC loss. Through experiments, the proposed method appears to perform comparably to the best-performing prior offline RL approaches despite being simple and requiring lower tuning complexity.

**Limitations And Societal Impact:**

Yes

**Main Review:**

The paper is well written and easy to understand. The proposed approach focuses on simplicity yet it seems to perform comparably to baselines in the empirical evaluations. The proposed method is also easier to tune and therefore makes it less costly to evaluate the train policies online after offline training. I think the method is of reasonable significance to the field of offline RL.

However, I do have a few concerns, which I will discuss as follows.

First, it seems that the novelty of the method is a bit limited. The authors seem to directly adapt RL+BC to the offline setting except that they add the state normalization, which is also not new. The authors also didn't theoretically justify the approach. For example, the authors should show that the method can guarantee safe policy improvement and moreover enjoys comparable or better policy improvement guarantees w.r.t. prior approaches. Without the theoretical justification and given the current form of the method, I think the method is a bit incremental.

Moreover, the empirical evaluations are not thorough. The authors only evaluated the approach in simple mujoco environments in D4RL. It is unclear if the method can perform well more undirected multitask datasets such as antmaze and kitchen as well as more complicated manipulation tasks such as adroit in D4RL. It also seems that the method does not perform well on random datasets. Is this a major limitation? I also think the authors should add state normalization to all baselines to ensure fair comparison as the state normalization is not a new technique in RL.

Finally, I think the comparisons are not complete. The authors should also compare the method to more recent model-free offline RL approaches such as [1] and model-based methods such as [2,3], which attains better performance on random and medium-replay datasets.

Overall, given the above comments, I would vote for a weak reject.

[1] Sinha, S., & Garg, A. (2021). S4RL: Surprisingly Simple Self-Supervision for Offline Reinforcement Learning. arXiv preprint arXiv:2103.06326.

[2] R. Kidambi, A. Rajeswaran, P. Netrapalli, and T. Joachims. Morel: Model-based offline reinforcement learning. arXiv preprint arXiv:2005.05951, 2020.

[3] T. Yu, G. Thomas, L. Yu, S. Ermon, J. Zou, S. Levine, C. Finn, and T. Ma. Mopo: Model-based offline policy optimization. arXiv preprint arXiv:2005.13239, 2020.


**Time Spent Reviewing:**

3 hours

---

> ### Author Response · Authors · 2021-08-10
> **Response to Reviewer 63kT**
>
> Thank you for your review & feedback.
>
> On novelty: We don’t disagree at all that our algorithm is incremental in novelty (we highlight a number of similar algorithms in the related work). However, our main claim/contribution is not so much that this is the best possible offline RL algorithm, or that it is particularly novel, but rather the surprising observation that the use of very simple techniques can match/outperform current algorithms. The hope is that TD3+BC could be used as an easy-to-implement baseline or starting point for other additions (such as S4RL), while eliminating a lot of unnecessary complexity, hyperparameter tuning, or computational cost, required by more sophisticated methods.
>
> On empirical evaluation: To the best of our knowledge, our strongest baseline, Fisher-BRC, is considered SOTA for model-free algorithms, and was published recently at ICML.
>
> Due to the standardization of D4RL results, we can compare directly with the suggested baselines (and we will include these results into the final draft). We report these below, but we would like to remark two points:
> 1.	MOReL and MOPO are from a different family of approaches (model-based) and both use environment-specific hyperparameters.
> 2.	S4RL is tangential to our approach and could very easily be added to our method by simply swapping CQL for TD3+BC. Our approach is arguably more amenable to these types of additions to a base algorithm since there are fewer hyperparameters, meaning we don’t have to worry about as many interactions between changes.
>
> |               |             | CQL + S4RL | MOReL | MOPO | TD3+BC |
> |---------------|-------------|------------|-------|------|--------|
> | Random        | HalfCheetah | 53.9       | 25.6  | 35.4 | 10.2   |
> |               | Hopper      | 10.7       | 53.6  | 11.7 | 11     |
> |               | Walker2d    | 25.1       | 37.3  | 13.6 | 1.4    |
> |||||||
> | Medium        | HalfCheetah | 48.6       | 42.1  | 42.3 | 42.8   |
> |               | Hopper      | 81.3       | 95.4  | 28   | 99.5   |
> |               | Walker2d    | 93.1       | 77.8  | 17.8 | 79.7   |
> |||||||
> | Medium Replay | HalfCheetah | 51.7       | 40.2  | 53.1 | 43.3   |
> |               | Hopper      | 36.8       | 93.6  | 67.5 | 31.4   |
> |               | Walker2d    | 35         | 49.8  | 39   | 25.2   |
> |||||||
> | Medium Expert | HalfCheetah | 78.1       | 53.3  | 63.3 | 97.9   |
> |               | Hopper      | 117.9      | 108.7 | 23.7 | 112.2  |
> |               | Walker2d    | 107.1      | 95.6  | 44.6 | 101.1  |
> |||||||
> | Expert        | HalfCheetah | -          | -     | -    | 105.7  |
> |               | Hopper      | -          | -     | -    | 112.2  |
> |               | Walker2d    | -          | -     | -    | 105.7  |
>
> Overall, we would argue that our method remains very competitive while providing a much higher degree of simplicity.
>
> Additionally, we will add state normalization results to the baselines (CQL, Fisher-BRC).
>
> |               |             | CQL   |        CQL + norm | Fisher-BRC |        Fisher-BRC + norm |
> |---------------|-------------|-------|-----------------|-----------|------------------------|
> | Random        | HalfCheetah | 21.7  | 21.5            | 32.2      | 31.5                   |
> |               | Hopper      | 10.7  | 10.6            | 11.4      | 11.1                   |
> |               | Walker2d    | 2.7   | 0.7             | 0.6       | 0.8                    |
> |||||||
> | Medium        | HalfCheetah | 37.2  | 37.8            | 41.3      | 41.3                   |
> |               | Hopper      | 44.2  | 46.7            | 99.4      | 99.3                   |
> |               | Walker2d    | 57.5  | 58.8            | 79.5      | 78.8                   |
> |||||||
> | Medium Replay | HalfCheetah | 41.9  | 41.9            | 43.3      | 43.7                   |
> |               | Hopper      | 28.6  | 29.9            | 35.6      | 0.7                    |
> |               | Walker2d    | 15.8  | 16.4            | 42.6      | 40.6                   |
> |||||||
> | Medium Expert | HalfCheetah | 27.1  | 32.1            | 96.1      | 90.2                   |
> |               | Hopper      | 111.4 | 112             | 90.6      | 90.1                   |
> |               | Walker2d    | 68.1  | 70.5            | 103.6     | 103.7                  |
> |||||||
> | Expert        | HalfCheetah | 82.4  | 98.3            | 106.8     | 107.9                  |
> |               | Hopper      | 111.2 | 112.1           | 112.3     | 112.4                  |
> |               | Walker2d    | 103.8 | 97.7            | 79.9      | 94.3                   |
>
> Ultimately, we don’t find the addition of state normalization provides the same level of benefit to the baselines, which might be because these methods would require hyperparameter tuning to compensate for an additional modification.

---

### Official Review · Reviewer_RqWM · 2021-07-23

**Rating:** 8
**Confidence:** 4

**Summary:**

This paper diagnoses some pitfalls with complex models in offline RL and proposes a very simple baseline model which attains competitive performance. This minimalist approach involves adding a behavior cloning term to the TD3 objective and normalizing the states and makes its hyperparameter easier to tune by normalizing by the value.

**Limitations And Societal Impact:**

Yes

**Main Review:**

This paper attempts to cut through the often-confusing tangle of current algorithms for offline RL by providing a more straightforward alternative. The proposed method is in fact simple and easy to implement. The scaling rule for its hyperparameter is the least intuitive component but remains much simpler than the methods it compares to.

Ablating the state of the art algorithms by removing their implementation adjustments is a sensible thing to do. This emphasizes the need for "test environments" which researchers do not have access to for hyperparameter tuning; this is especially pressing in offline RL as access to the environment should be considered expensive and methods may overfit to the specific benchmark dataset. This need remains future work.

Overall this is the kind of paper that I'm glad someone wrote. It does not have sensational results or a flashy method, but it points out how much of current offline RL work is light with no heat. It provides a baseline that is easy and fast to train and which should probably be the bar for new research going forward.

### Notes to the authors
* Missing "which" or other connector at the end of line 188.

**Time Spent Reviewing:**

3

---

> ### Author Response · Authors · 2021-08-10
> **Response to Reviewer RqWM**
>
> Thank you for your review and positive comments. We appreciate how you’ve highlighted the significance of our work in the space of current offline RL research.
>
> We will correct the error on line 188, thank you.

---

### Decision · Program_Chairs · 2021-09-27

**Decision:**

Accept (Spotlight)

**Comment:**

Overall this paper makes a nice contribution and is arguably an example of the type of work our fields need more of.

My only reservation is that the authors should have done more to highlight deficiencies of the approach, which may not be present for some of the other approaches. In particular, the approach obviously should suffer when data has coverage of good policies, but may not correspond to a good policy that one should aim to imitate. The authors are aware of this and will ideally make this point more strongly in the revision. Regardless, the paper still makes a nice contribution.